# Cellular Immune Responses to SARS-CoV-2 in Exposed Seronegative Individuals

**DOI:** 10.3390/v15040996

**Published:** 2023-04-18

**Authors:** Natasha J. Norton, Kayla A. Holder, Danielle P. Ings, Debbie O. A. Harnum, Rodney S. Russell, Michael D. Grant

**Affiliations:** 1Immunology and Infectious Diseases Program, Division of BioMedical Sciences, Faculty of Medicine, Memorial University of Newfoundland and Labrador, St. John’s, NL A1B 3V6, Canada; 2Eastern Regional Health Authority, Department of Health and Community Services, St. John’s, NL A1B 3V6, Canada

**Keywords:** SARS-CoV-2, exposed uninfected, cross-reactivity, OC43, HKU1, cellular immunity

## Abstract

Some SARS-CoV-2-exposed individuals develop immunity without overt infection. We identified 11 individuals who were negative by nucleic acid testing during prolonged close contact and with no serological diagnosis of infection. As this could reflect natural immunity, cross-reactive immunity from previous coronavirus exposure, abortive infection due to de novo immune responses, or other factors, our objective was to characterize immunity against SARS-CoV-2 in these individuals. Blood was processed into plasma and peripheral blood mononuclear cells (PBMC) and screened for IgG, IgA, and IgM antibodies (Ab) against SARS-CoV-2 and common β-coronaviruses OC43 and HKU1. Receptor blocking activity and interferon-alpha (IFN-α) in plasma were also measured. Circulating T cells against SARS-CoV-2 were enumerated and CD4^+^ and CD8^+^ T cell responses discriminated after in vitro stimulation. Exposed uninfected individuals were seronegative against SARS-CoV-2 spike (S) and selectively reactive against OC43 nucleocapsid protein (N), suggesting common β-coronavirus exposure induced Ab cross-reactive against SARS-CoV-2 N. There was no evidence of protection from circulating angiotensin-converting enzyme (ACE2) or IFN-α. Six individuals had T cell responses against SARS-CoV-2, with four involving CD4^+^ and CD8^+^ T cells. We found no evidence of protection from SARS-CoV-2 through innate immunity or immunity induced by common β-coronaviruses. Cellular immune responses against SARS-CoV-2 were associated with time since exposure, suggesting that rapid cellular responses may contain SARS-CoV-2 infection below the thresholds required for a humoral response.

## 1. Introduction

Since its introduction into the human population in late 2019, severe acute respiratory syndrome coronavirus 2 (SARS-CoV-2) has spread widely and continues to circulate globally. As of March 2023, there have been over 675 million known cases of coronavirus disease (COVID-19) worldwide and almost 7 million related deaths. Canada alone has had over 4.5 million documented cases and nearly 50,000 related deaths (John Hopkins Coronavirus Resource Center). It is estimated that more than 75% of the Canadian population has now been infected with SARS-CoV-2. Research over the last three years has increased the understanding of the virus and host immune response to infection and has helped to inform public health agencies on best practices to address the pandemic. As with other viruses, exposure to SARS-CoV-2 can occasionally occur with no overt signs of infection, negative nucleic acid-based testing, and no subsequent seroconversion. In a small fraction of cases, this is associated with detectable cellular immunity against SARS-CoV-2. Recent studies have explored the incidence of abortive infections, which occur when virally infected cells produce no progeny virus following exposure to SARS-CoV-2. A cohort of seronegative healthcare workers in the United Kingdom who were tested during the initial wave of COVID-19 (March 2020) had evidence of SARS-CoV-2-specific T cell responses [1]. A similar phenomenon of specific T cell responses in the absence of seroconversion was previously documented with exposure to hepatitis C virus (HCV) [2,3]. This suggests that in rare cases, viral infections can be curtailed prior to seroconversion by either pre-existing or rapidly developing cellular immunity.

Pre-existing cellular immunity against SARS-CoV-2 could result from exposure to common coronaviruses that share T cell epitopes with SARS-CoV-2 [4,5,6]. Antibodies induced by circulating endemic α- and β-coronaviruses, NL63 and 229E, and OC43 and HKU1, respectively, which typically cause mild respiratory illness [7], cross-react with SARS-CoV-2 proteins [8,9,10]. Common coronavirus antibody cross-reactivity was also noted during the SARS-CoV-1 outbreak [11]. Infection with a common coronavirus prior to infection with SARS-CoV-2 can lessen COVID-19 disease severity [12]; however, it is unclear whether cross-reactive antibodies or other forms of immunity induced by infection with common coronaviruses provide protection against SARS-CoV-2 infection or against severe COVID-19 [8,9,12].

In this study, we investigated immune responses against SARS-CoV-2 of individuals who were in prolonged close contact to an active case of COVID-19 yet were seemingly uninfected. These individuals showed no evidence of viral replication by reverse transcriptase (RT) polymerase chain reaction (PCR) testing, had no self-reported symptoms, and remained seronegative against the immunodominant SARS-CoV-2 spike (S) protein.

## 2. Materials and Methods

### 2.1. Selection of Study Participants and Sample Collection

This study was approved by the Newfoundland and Labrador Health Research Ethics Authority and carried out in accordance with the recommendations of the Canadian Tri-Council Policy Statement: Ethical Conduct for Research Involving Humans. Study subjects are nested within a cohort established for an ongoing study at the Memorial University of Newfoundland and Labrador, where 263 participants were recruited based on previous RT PCR-confirmed or suspected SARS-CoV-2 infection [13]. Written informed consent was obtained for whole blood collection in accordance with the Declaration of Helsinki and subjects completed a questionnaire at study intake on SARS-CoV-2 exposure, testing, and symptom history. Through purposive sampling, individuals who reported close prolonged contact, either through a spouse or family member, to an active case of SARS-CoV-2 yet did not test positive for COVID-19 via PCR, were selected for further testing. Prolonged close contact included such things as caring for a partner throughout their illness, shared sleeping arrangements, shared eating and washroom facilities, exercise partners, ride sharing, and household proximity throughout the presumed infectious period of 5 days or more. All subjects identified as fitting this criterion were included. This was designed as an observational study without sample size calculation to assess the requirement for a valid estimate of overall frequency of such cases in the population. Whole blood was drawn by forearm venipuncture into acid-citrate-dextrose preserved vacutainers and plasma was collected after centrifuging whole blood for 10 min at 500 g. Plasma was stored immediately in small aliquots at −80 °C until analysis. Peripheral blood mononuclear cells (PBMC) were isolated from the cellular fraction of blood following the consensus protocol established by the Canadian Autoimmunity Standardization Core procedure [14]. Isolated PBMC were cryopreserved in 10% DMSO (Sigma-Aldrich, St. Louis, MO, USA), 90% fetal bovine serum (FBS, HyClone™, GE Healthcare Life Sciences, Logan, UT, USA) at ≤2.0 × 10^7^/mL by cooling to −80 °C in a Frosty^TM^ freezing container overnight before transfer to LN_2_.

### 2.2. Serological Testing

Plasma was diluted in phosphate buffered saline (PBS) containing 0.05% Tween 20 (Fisher Bioreagents, Thermo Fisher Scientific, Rochester, NY, USA) and 0.1% bovine serum albumin (BSA, Sigma-Aldrich, St. Louis, MO, USA) and tested for anti-SARS-CoV-2 antibodies by enzyme-linked immunosorbent assay (ELISA) using recombinant proteins as antigens. Proteins were coated overnight (4 °C) at 50 ng/well in 50 μL Dulbecco’s PBS (Corning, Mediatech, Inc., Manassa, VA, USA) onto 96-well Immununlon-2 HB (Thermo Fisher Scientific, Waltham, MA, USA) ELISA plates to test for antibodies against SARS-CoV-2 Wuhan-Hu-1 receptor binding domain (RBD; Sino Biological, Wayne, PA, USA), nucleocapsid (N) proteins (Sino Biological) and full-length spike (FLS, SMT1-1 National Research Council of Canada), and β-coronavirus N proteins from OC43 and HKU1 (Sino Biological). Plates were washed 4 times after coating and 6 times between all subsequent steps with 300 μL/well PBS + 0.05% Tween^®^ 20. Plates were blocked with 200 μL 1% BSA in PBS for 1 h, after which 100 μL diluted plasma was added for 1.5 h, and 100 μL diluted goat-anti human IgG, IgA, or IgM horseradish peroxidase (HRP) conjugated detection antibodies (IgG and IgA Jackson ImmunoResearch, Baltimore Pike, West Grove, PA, USA; IgM NCI Biological Resources Branch, Frederick National Library, Fredrick, MD, USA) were added to the wells for 1 h. Colour was developed using 100 μL of 3,3′,5,5′-tetramethylbenzidine (TMB, T8665, Sigma-Aldrich, St. Louis, MO, USA) for 20 min and the reaction was stopped by adding 100 μL of 1 M H_2_SO_4_. Optical density (OD) was read at 450 nm on a BioTek Synergy HT plate reader. Plasma was diluted 1:100 to test for IgG antibodies and 1:50 to test for IgM and IgA antibodies. The anti-IgG*horseradish peroxidase (HRP) and anti-IgM*HRP conjugates were diluted 1:50,000 and 1:25,000, respectively. A set of 40 control serum samples collected before October 2019 was used to establish cut-off OD values for IgG seropositivity against SARS-CoV-2 S and RBD [13]. Any sample producing an OD more than 2 standard deviations (SD) above the mean OD of the 40 control samples was considered seropositive.

### 2.3. SARS-CoV-2 Pseudo-Neutralization ELISA

Immulon-2 96 well ELISA plates were coated with 100 ng of SARS-CoV-2 FLS protein in 50 μL PBS overnight at 4 °C. The plates were then washed 4 times with 300 μL/well of PBS + 0.05% Tween 20, blocked with 200 μL 1% BSA in PBS for 1 h, and then washed 4 more times. Plasma was diluted 1:100 in 0.1% BSA in PBS + 0.05% Tween 20 (diluent) and 100 μL was added to the respective wells for 1.5 h, then washed 6 times. Biotinylated ACE2 (RayBiotech Life, Inc., Peachtree Corners, GA, USA), made up in diluent, was added in 100 μL at 40 ng/well for 1 h and then the plate was washed 6 times. Next, streptavidin (SA)-HRP, Jackson ImmunoResearch, Baltimore Pike, West Grove, PA, USA) diluted 1:50,000 was added to the wells for 1 h and the plates were washed another 6 times. The enzymatic colour reaction was developed using 100 μL TMB per well for 20 min and stopped with 100 μL 1 M H_2_SO_4_. Optical density was read at 450 nm on a BioTek Synergy HT plate reader. Percent neutralization was calculated using the following equation:(1)% Inhibition=(1−OD450 nm of sampleOD450 nm of negative control×100

### 2.4. Measurement of Interferon-α

Plasma interferon alpha (IFN-α) levels were measured using the RayBio^®^ Human IFN-α kit (RayBiotech Life, Inc., Peachtree Corners, GA, USA) following the manufacturer’s instructions. Briefly, all reagents, standards, and samples were brought to room temperature before use. Samples and standards (100 μL) were added to the respective wells for 2.5 h with gentle shaking at room temperature. After 4 300 μL washes, 100 μL of biotinylated detection antibody was added to wells for 1 h with gentle shaking, then plates were washed. Next, 100 μL SA-HRP was added for 45 min, again with gentle shaking. Following another wash, 100 μL TMB was added, colour developed for 20 min in the dark, and the reaction stopped with 50 μL stop solution. Plates were read at 450 nm on a BioTek Synergy HT plate reader. The calculation of the IFN-α concentration in samples was performed based on the standard curve generated.

### 2.5. ELISpot Assay

Peripheral blood mononuclear cells were recovered by rapid thawing in a 37 °C water bath and added to 9 mL of lymphocyte medium (LM; RPMI-1640, 10% FBS [HyClone™], 200 IU/mL penicillin/streptomycin [Invitrogen, Thermo Fisher Scientific, Waltham, MA, USA], 0.01 M HEPES [Invitrogen], and 2 × 10^−5^ M 2-mercaptoethanol [Sigma-Aldrich]). Cells were then centrifuged at 450× *g* for 5 min, resuspended in 5 mL of CTL medium (CTL-Test™ Medium, CTL ImmunoSpot^®^, Shaker Heights, OH, USA) with 1% l-glutamate, and maintained overnight at 37 °C, 5% CO_2_ prior to use on ELISpot. Recovered cells were added in duplicate at 2.0 × 10^5^/well to 96-well pre-coated IFN-γ ELISpot plates (CTL ImmunoSpot^®^). The cells were stimulated with peptide pools of SARS-CoV-2 spike, nucleocapsid, membrane, and envelope proteins with the final concentration of each peptide at 1 μg/mL. PepTivator^®^ SARS-CoV-2 ProtS complete peptide pool mainly consisted of 15-mer amino acid (aa) sequences with 11 aa overlaps (Miltenyi Biotec, San Diego, CA, USA) resuspended in endotoxin-free ultra-pure water (H_2_O, Millipore, Sigma-Aldrich). All other SARS-CoV-2 protein peptide pools were pooled from BEI resources peptide sets (Nucleocapsid NR-52419, Envelope NR-52405, Membrane NR-52403) and consisted of 17-mer aa sequences with 10–11 aa overlap to cover the whole protein of interest resuspended in dimethyl sulfoxide (DMSO) and diluted in unsupplemented RPMI 1640. The membrane and envelope peptides were combined into a single pool. Cells and peptides, along with their respective vehicle controls (H_2_O or DMSO) and anti-CD3 as a positive control (OKT3, ATCC, CRL-8001), were incubated for 24 h at 37 °C, 5% CO_2_. After 24 h, the plate was washed twice with 200 μL/well PBS and another 2 times with 200 μL/well PBS + 0.05% Tween 20. Anti-human IFN-γ detection antibody was diluted in diluent B, filtered through a 0.1 μm filter, and 80 μL was added to each well. Following a 2 h incubation at room temperature, the plate was washed 3 times with 200 μL/well PBS + 0.05% Tween 20. Next, 80 μL of a tertiary solution (SA-HRP) in diluent B was added to wells for 30 min at room temperature, then plates were washed twice with 200 μL/well PBS + 0.05% Tween 20 and then two more times with 200 μL/well dH_2_O. Colour was developed using 80 μL of developer solution in the dark at room temperature for 15 min and the reaction was stopped by gently rinsing the plate with tap water 3 times. The plate was airdried overnight, then scanned and counted on a CTL ImmunoSpot^®^ S6 Universal Analyzer (CTL Analyzers, Shaker Heights, OH, USA). Subjects who had ≥50 IFN-γ producing T cells/10^6^ PBMC above the vehicle control background following stimulation with at least one of the peptide pools were considered to have a specific cellular immune response against SARS-CoV-2. Results are shown with the vehicle control background subtracted.

### 2.6. In Vitro Stimulation with SARS-CoV-2 Peptides

Recovered cells from samples used in ELISpot assays that yielded ≥50 IFN-γ producing T cells/10^6^ PBMC in response to at least one of the peptide pools were then stimulated for 7 days with the same peptide pool(s) in vitro as previously described [15]. Depending on availability, from 2 × 10^6^ to 5 × 10^6^ total PBMC were pelleted and stimulated in small volumes for 1 h at 37 °C, 5% CO_2_ with SARS-CoV-2 spike, nucleocapsid, or membrane and envelope combination peptide pools (1 μg each individual peptide). After 1 h, the culture volume was increased to 1 mL using LM supplemented with 25 ng/mL interleukin (IL)-7 (National Cancer Institute, Frederick, MD, USA). These cells were then incubated for 7 days at 37 °C, 5% CO_2_, adding LM when needed to support their growth.

### 2.7. Flow Cytometry

After 7 days’ in vitro stimulation at 37 °C, 5% CO_2_, responder cells were analyzed by short-term restimulation and flow cytometry as previously described [15]. Briefly, 5.0 × 10^5^ cells were restimulated in a final volume of 500 μL with the SARS-CoV-2 peptide pool of interest at 1 μg/mL for each peptide or a matching volume of vehicle control, with Brefeldin A (Sigma-Aldrich) added to a final concentration of 10 μg/mL. After 5 h, the cells were washed with flow cytometry buffer (1 X PBS, 5 mM EDTA, 0.2% NaN_3_, 0.5% FBS) and stained with the following fluorochrome conjugated antibodies for 20 min in the dark: αCD3 (VioGreen™, REAfinity™ Clone REA613, Miltenyi Biotec), αCD4 (APC-Vio^®^ 770, REAfinity™ Clone REA623, Miltenyi Biotec), and αCD8 (PerCP, Clone HIT8a, BioLegend, San Diego, CA, USA). The cells were washed again using flow cytometry buffer and intracellular IFN-γ stained following the MACS Miltenyi Biotec intracellular staining of eukaryotic cells procedure and kit. Briefly, the cells were fixed in a final volume of 500 μL using equal amounts of Inside Fix and buffer (PBS pH 7.2, 0.5% BSA, and 2 mM EDTA) for 20 min in the dark, centrifuged, then washed using flow cytometry buffer. Next, anti-IFN-γ (PE, eBioscience™ Clone 4S.B3, Invitrogen, Thermo Fisher Scientific), diluted in Inside Perm to a final volume of 100 μL, was added and incubated at room temperature for 10 min. Next, 1 mL of Inside Perm was added to each sample and the samples were centrifuged, decanted, and resuspended in the remaining liquid prior to analysis on a Beckman Coulter CytoFLEX flow cytometer (Beckman Coulter, Brea, CA, USA). At least 100,000 events were collected for each sample stimulation condition. We gated on PBMC, distinguished CD3^+^ cells and then gated separately on CD3^+^CD4^+^ and CD3^+^CD8^+^ cells to analyze IFN-γ expression by each T cell subset. The background from unstimulated conditions was subtracted from the percentage of IFN-γ producing cells in test conditions to calculate the percentage of CD4^+^ or CD8^+^ T cells producing IFN-γ in response to SARS-CoV-2 peptides. Data were analyzed and visualized using Kaluza Version 2.1 (Beckman Coulter).

### 2.8. Statistical Analysis

All statistical analyses were conducted using GraphPad Prism Version 9.5.0. Significance values, where applicable, are shown above lines spanning the groups compared. The following statistical tests were conducted for data analysis in this study, as specified in the relevant figure captions: Mann–Whitney test; Wilcoxon signed rank test; Spearman correlation.

## 3. Results

### 3.1. Selection of SARS-CoV-2 Exposed Uninfected Persons

For a study initiated in March 2020, we recruited individuals with confirmed COVID-19, suspected COVID-19, and contacts of persons with confirmed COVID-19 into a study of immune responses against SARS-CoV-2. Prior to the widespread introduction of COVID-19 vaccines, we identified 11 non-immunocompromised individuals (Table 1) defined as discordant cases who were in prolonged close contact through their spouse or family member(s) with active cases of COVID-19. Of these, 7 were exposed to the ancestral Wuhan-Hu-1 strain of SARS-CoV-2 between 15 March and 4 April 2020, and 4 were exposed to the SARS-CoV-2 Alpha variant (B.1.1.7) between 10 February and 15 February 2021. Despite prolonged close contact with one or more confirmed cases of COVID-19, these 11 exposed individuals had negative RT-PCR test results at the time and reported no symptoms of infection throughout or shortly after the course of their close contact. No other exposures or signs of infection were noted before their sample collection dates. All samples used in this study were collected prior to any instance of COVID-19 vaccination, COVID-19 infection, or documented infection with another coronavirus.

### 3.2. Anti-SARS-CoV-2 Serology

Although infection with SARS-CoV-2 can occur without seroconversion, especially in mild or asymptomatic cases, we investigated antibody responses against SARS-CoV-2 in the discordant individuals to corroborate the absence of overt infection indicated by negative PCR tests. All 11 discordant individuals were seronegative for IgG antibodies against SARS-CoV-2 RBD and FLS (Figure 1a). All discordant individuals were also seronegative for IgM antibodies against FLS (Figure 1b). Several studies reported the detection of IgA antibodies against SARS-CoV-2 S either before, or in the absence of IgG antibodies, therefore, we also measured IgA antibodies against FLS. All discordant individuals were seronegative for IgA antibodies against FLS (Figure 1c). To test the possibility that the immune system of the discordant case individuals was primed to respond to vaccination, similar to what occurred in previously infected individuals, we compared the IgG anti-S response after one dose of the Pfizer BioNTech (BNT162b2) mRNA vaccine of 5 discordant cases for whom we had post-vaccination samples to that of age, sex, and days-post-vaccination-matched previously infected individuals and non-exposed individuals (Figure 1d). The discordant cases had an IgG anti-FLS antibody response to vaccination similar to non-exposed individuals, while the previously infected individuals had a significantly greater IgG response. Thus, we found no evidence of a humoral response against SARS-CoV-2 S in the exposed uninfected individuals, nor of occult priming for a humoral response to S-based SARS-CoV-2 vaccination.

### 3.3. Innate Immunity against SARS-CoV-2

To investigate whether any of these discordant individuals had evidence of innate immunity against SARS-CoV-2, we tested for receptor blocking activity and measured IFN-α in their plasma. There was no significant plasma-mediated inhibition of SARS-CoV-2 S binding to angiotensin-converting enzyme 2 (ACE2) in any of the 11 discordant cases (Figure 2a) and no significant difference in circulating IFN-α levels for the 11 discordant individuals compared to matched controls (Figure 2b).

### 3.4. Cross-Reactive Immunity with Common β-Coronaviruses

In a previous study, we found that some individuals who were seronegative for antibodies against SARS-CoV-2 S have cross-reactive antibodies against SARS-CoV-2 N protein resulting from infection with common β-coronaviruses [1]. To investigate cross-reactive immunity against common β-coronaviruses in the discordant individuals, we measured plasma IgG anti-N antibodies against SARS-CoV-2, OC43, and HKU1 β-coronaviruses. Relatively low but detectable levels of IgG reactivity against SARS-CoV-2 N protein were present in plasma from the 11 discordant individuals (Figure 3). Antibody activity was significantly greater against HKU1 and OC43 N proteins compared to SARS-CoV-2, suggesting that previous infection with these common β-coronaviruses underlay the presence of antibodies against SARS-CoV-2 N protein. Since exposure to the common β-coronaviruses was ubiquitous prior to the emergence of COVID-19, we compared IgG anti-SARS-CoV-2, OC43, and HKU1 N levels in the 11 discordant individuals’ plasma samples to levels in plasma from a set of age- and sex-matched individuals collected before October 2019. There was no significant difference in anti-N antibody levels against any N protein between the groups (Figure 3), indicating that the development of cross-reactive antibodies against SARS-CoV-2 N from previous exposure to common β-coronaviruses was not a distinguishing feature of the 11 discordant individuals whom we identified.

### 3.5. T Cell Responses to SARS-CoV-2

Antigen-specific T cell responses have previously been reported in individuals who tested negative by PCR and remained seronegative following exposure to SARS-CoV-2 proteins. We tested for cellular immune responses in our discordant case cohort using three peptide pools spanning SARS-CoV-2 S, N, and envelope/membrane (E/M) proteins. Six of the eleven discordant individuals tested had ≥50 IFN-γ producing T cells/10^6^ PBMC above background in response to at least one of the peptide pools and were deemed responders on this basis (Figure 4a). Of note, subject 1185 had the greatest response with 475 IFN-γ producing S-specific T cells/10^6^ PBMC and subject 1637 had greater than 100 IFN-γ producing T cells/10^6^ PBMC in response to all 3 peptide pool stimulations. Thus, more than half of the discordant cases showed evidence of T cell immunity, either from previous infection with common coronaviruses or from exposure to SARS-CoV-2 through close personal contact with one or more infected family members. The magnitude of the IFN-γ responses correlated significantly with time between sample collection and SARS-CoV-2 exposure (Figure 4c). A comparison of time between exposure and sample collection for ELISpot responders versus non-responders indicated that the responders as a group had significantly less time between SARS-CoV-2 exposure and sample collection (Figure 4c). While the T cell responses could reflect responses to cross-reactive epitopes in common coronaviruses, the inverse correlation between time since exposure and magnitude of the T cell response suggests a specific cellular response induced by exposure to SARS-CoV-2. This relationship with time since exposure also suggests that additional cellular immune responses against SARS-CoV-2 in the exposed seronegative individuals might have been detected if we had collected samples at earlier time points post exposure. Time since exposure was the only parameter that significantly correlated with the strength of the cellular immune response against SARS-CoV-2. No other immunological parameters measured correlated significantly, including IgG responses against coronavirus N proteins, and IgG, IgA, and IgM responses against SARS-CoV-2 S protein. Age can also play a role in reducing the strength of immune responses induced following exposure to antigens and reducing their durability, but we found no significant difference in age between responders and non-responders (Figure 4d).

To discriminate CD4^+^ and CD8^+^ T cell responses against SARS-CoV-2 in these individuals, the PBMC of responders from the same sample time point used for ELISpot assays were stimulated with SARS-CoV-2 peptide pools for 7 days’ in vitro and then analyzed for IFN-γ production following 5 h restimulation (Figure 5a–f). Of the 6 responders by ELISpot, 4 had both CD4^+^ and CD8^+^ T cell responses to the SARS-CoV-2 peptide pools following in vitro restimulation (Figure 5g,h). Notably, subject 1185 had 4.8% of their CD8^+^ T cell population responding against S and subject 1638 had 8.7% of their CD8^+^ T cell population responding against the E/M peptide pool combination. There were 4 individuals in our discordant cohort who had robust SARS-CoV-2-specific T cell responses between 2 and 6 months following exposure despite testing PCR negative, remaining seronegative, and having no overt symptoms of infection. This may indicate that these individuals experienced viral replication but cleared all viral progeny before seroconversion through either rapidly developing immunity or pre-existing cross-reactive immunity.

## 4. Discussion

In this this study, we assessed immunity against SARS-CoV-2 in individuals who experienced prolonged close contact with COVID-19 but showed no overt signs of infection. As these samples were collected prior to the widespread availability of COVID-19 vaccines, this cohort of discordant cases all lacked detectable IgG responses to SARS-CoV-2 S protein. Seronegativity for SARS-CoV-2 S clearly distinguished them from their infected close contacts and from the vast majority of individuals with RT-PCR-confirmed SARS-CoV-2 infection. Within the discordant cohort, there was also no evidence of IgM or IgA antibodies specific for SARS-CoV-2 S protein. Anti-SARS-CoV-2 N IgG responses were present within the cohort, but by comparing them with IgG responses against two common β-coronavirus (OC43 and HKU1) N proteins, we concluded that the response against SARS-CoV-2 most likely represented cross-reactive antibodies due to previous immunogenic exposure(s) to common β-coronaviruses. Similar results showing cross-reactive antibodies against SARS-CoV-2 from common coronavirus infection have previously been reported [9,10]. Comparison to pre-pandemic IgG antibody responses against SARS-CoV-2, OC43, and HKU1 in age- and sex-matched controls revealed that cross-reactive antibodies against SARS-CoV-2 N were no more prominent in the discordant case cohort than in the general population. While we can infer that common β-coronaviruses, specifically OC43, circulated in Newfoundland and Labrador and induced antibodies cross-reactive against SARS-CoV-2, there was no evidence that more repeated or more recent exposure was responsible for an apparent resistance to SARS-CoV-2 infection in the exposed uninfected individuals whom we identified. Given their widespread circulation, especially amongst younger, school-aged individuals, we would expect the majority of the population has been exposed to common coronaviruses and, thus, have circulating IgG antibodies cross-reactive against SARS-CoV-2 N protein.

A previous suggestion that elevated levels of circulating ACE2 provide some protection from severe COVID-19 [16] led us to consider the possibility that higher-than-normal levels of circulating ACE2 could also protect against infection by inhibiting SARS-CoV-2 attachment to membrane-bound ACE2. However, within this cohort, we saw no evidence of meaningful receptor blocking by circulating ACE2. Constitutively elevated levels of circulating IFN-α can provide non-specific protection from viral infection [17], but there was no evidence of higher plasma IFN-α levels in the discordant cohort compared to matched control groups.

The detection of T cell responses against SARS-CoV-2 in seronegative individuals varies quite widely based on the method used, with proliferation and expression of markers of immune activation more licentious than IFN-γ ELISpot. We tested against peptides representing only a small fraction of the SARS-CoV-2 genome and 6 of the 11 discordant cases that we identified had circulating SARS-CoV-2-specific T cells detectable by IFN-γ ELISpot. The 5 exposed individuals whom we identified that did not have detectable T cell responses in our ELISpot assay were tested after a significantly longer interval from exposure than those with T cell responses, suggesting responses may have waned below detectable levels in some of these 5 cases. It is likely that the T cell responses elicited from exposure without seroconversion are less durable than those elicited by PCR-confirmed infection and especially less durable than those elicited by severe infection [15,18,19]. Due to the waning of responses, the true incidence of induction of cellular immunity against SARS-CoV-2 from exposure without seroconversion may be underestimated when testing is delayed. When testing only includes a small subset of potential T cell epitopes from SARS-CoV-2, this also increases the possibility of underestimating the incidence of cellular immunity.

Further investigation of the T cell response to SARS-CoV-2 following a 7-day in vitro stimulation showed that 4 of the 6 ELISpot responders within the discordant case cohort had both CD4^+^ and CD8^+^ T cell responses to the same SARS-CoV-2 peptide pools, indicative of some level of viral replication in host cells. This type of robust virus-specific T cell response in the absence of seroconversion has also been reported in some cases of asymptomatic COVID-19 [20], in populations at high risk for hepatitis C virus (HCV) infection, and in populations at high risk for human immunodeficiency virus (HIV) infection [21,22,23,24]. The 2 cases with SARS-CoV-2-specific T- cell responses detected on ELISpot with no evidence of in vitro expansion of SARS-CoV-2-specific T cells following the 7-day stimulation could be attributed to collateral activation by cytokines from T cells responding to a non-SARS-CoV-2 antigen or acute IFN-γ production by unstable memory T cells [24,25,26,27].

While the relationship between time since exposure and a detectable T cell response offers evidence that exposure to SARS-CoV-2 underlies these responses, several previous studies showed that SARS-CoV-2-specific T cell responses in exposed uninfected individuals can result from cross-reactive responses against epitopes shared with common β-coronaviruses [9,28]. Of 100 potential antigenic S peptides identified among SARS-CoV-2 and 4 common coronaviruses, NL63, 229E, OC43, and HKU1, 8 have ≥67% aa identity, indicating possible cross-reactivity [29]. Delineation of the specific epitopes that elicited responses by T cells in the exposed uninfected individuals and their comparison across SARS-CoV-2, OC43, and HKU1 could resolve this issue; however, we were limited by the availability of PBMC collected prior to vaccination from the 11 discordant case individuals. Given the relationship between detection of the responses and time since exposure to SARS-CoV-2 and similar examples with other viruses, we favour the possibility that the SARS-CoV-2-specific T cell response reflects an acute response to SARS-CoV-2 exposure rather than long-term cross-reactive T cell memory formed through previous infection with common β-coronaviruses. Considering the small sample size and other confounding factors, this remains a speculative assumption.

If responses observed in the exposed uninfected individuals reflect exposure to SARS-CoV-2 in a setting that allows de novo T cell responses in the absence of seroconversion, this likely relates to individual variability in the nature of exposure. Expansion of both CD4^+^ and CD8^+^ T cells in 4 of 6 responders following in vitro stimulation with SARS-CoV-2 peptides indicates exposure to low amounts of replicating virus that can stimulate T cell responses in the absence of seroconversion. This phenomenon was illustrated three decades ago in immunological studies following mucosal exposure to HIV and in simian immunodeficiency virus (SIV) vaccination experiments [30,31]. It is possible that within our discordant cohort, low levels of viral replication resulted in T cell memory development, but this short-lived acute infection was cleared by the T cells themselves or by other factors before a detectable antibody response could develop. There was no evidence that the humoral response to COVID-19 vaccination was primed through whatever exposure elicited the T cell responses. While this study was carried out with a small number of subjects, the fact that 11 discordant cases were identified within a relatively small number of subjects screened suggests that this is not a rare phenomenon. The overall significance of antiviral cellular immunity developing in the absence of seroconversion is unknown as it remains an open question whether its rapid development played a key role in viral containment and whether determining the fine specificity of these responses can inform better vaccine strategies. It will be important to investigate this phenomenon in larger, more controlled studies to determine if and how pre-existing or rapidly developing cellular immunity can abrogate SARS-CoV-2 infection.

This investigation was limited by the rarity of exposed uninfected individuals recruited into our study. In light of this small sample size, the results reported may not extend beyond the group studied and not apply to the general population. Conclusive results on the presence or absence of cellular immune responses against SARS-CoV-2 would require samples from earlier time points and testing against the entire SARS-CoV-2 peptidome, which was not possible due to limiting cell numbers and the availability of SARS-CoV-2 peptide sets. Therefore, we cannot definitively exclude the possibility of cellular immune responses against SARS-CoV-2 being present in the individuals categorized as non-responders based on our ELISpot assays. We were also limited in the ability to conduct confirmatory or follow-up testing by a lack of additional samples collected closer to the time of exposure and prior to vaccination or subsequent COVID-19 infection.

## Figures and Tables

**Figure 1 viruses-15-00996-f001:**
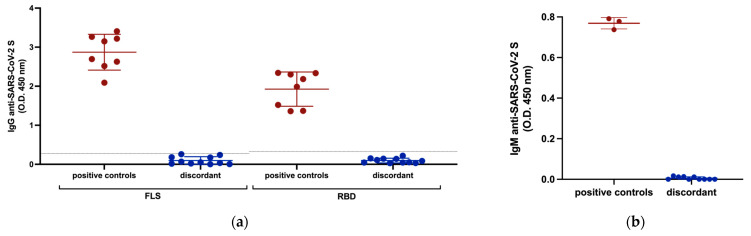
Serological responses of discordant case subjects against SARS-CoV-2 S measured by ELISA. (**a**) IgG antibody responses to SARS-CoV-2 FLS and RBD with plasma samples from previously infected subjects included as positive controls. The horizontal lines represent cut-off values for seropositivity established as 2 SD above the mean OD of control samples collected before October 2019. (**b**) IgM antibody response of discordant subjects with plasma samples from previously infected subjects included as positive controls against SARS-CoV-2 FLS. (**c**) IgA antibody response of discordant subjects with plasma samples from previously infected subjects included as positive controls against SARS-CoV-2 FLS and RBD. (**d**) Comparison of IgG antibody responses against SARS-CoV-2 FLS protein following one dose of BNT162b2 mRNA vaccination between previously infected, non-exposed, and 5 discordant individuals (1185, 1340, 1383, 1418, 1637) for whom post-vaccine 1 samples were available. Red and blue shading of dots and bars represents positive controls and discordant case subjects respectively. Black dots and the gray shaded bar represent previously unexposed subjects. The probability of a significant difference between groups was calculated using the Mann–Whitney test, with *p* values or ns (not significant) shown above lines spanning the groups compared.

**Figure 2 viruses-15-00996-f002:**
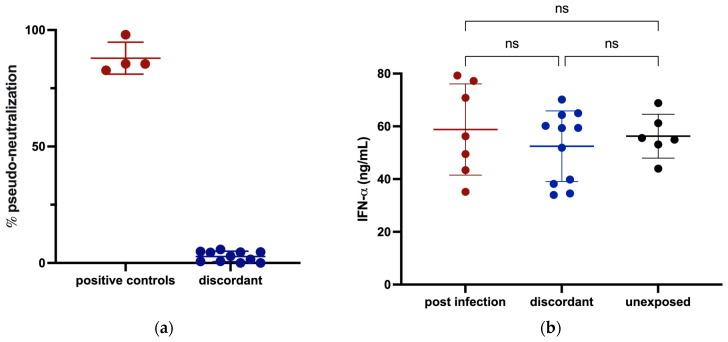
Assessment of potential for innate protection against SARS-CoV-2 infection measured by ELISA. (**a**) The ability of plasma from discordant subjects to inhibit the SARS-CoV-2 S interaction with ACE2 was tested with plasma from previously infected subjects included as positive controls. (**b**) Circulating IFN-α levels in plasma from discordant subjects and age- and sex-matched previously infected and unexposed individuals were measured and compared. Red, blue and black shading of dots represents positive controls, discordant case subjects and unexposed subjects respectively. The probability of a significant difference between groups was calculated using the Mann–Whitney test, with ns (not significant) shown above lines spanning the groups compared.

**Figure 3 viruses-15-00996-f003:**
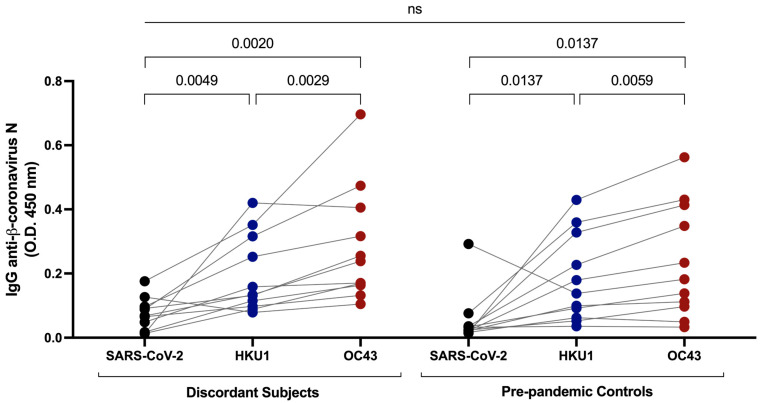
Measurement of cross-reactive antibodies against SARS-CoV-2 and common β-coronavirus N proteins by ELISA. IgG anti-β-coronavirus N antibodies against SARS-CoV-2, HKU1, and OC43 in plasma samples from the discordant cases were measured and compared to pre-pandemic plasma samples from age- and sex-matched controls. Black, blue and red shaded dots represent the IgG response against SARS-CoV-2, HKU1 and OC43 N proteins respectively. The probability of a significant difference between groups was calculated by Mann–Whitney test and responses to different N proteins compared by Wilcoxon signed rank test, with *p* values above lines spanning the groups compared. There was no significant difference between the discordant and pre-pandemic samples.

**Figure 4 viruses-15-00996-f004:**
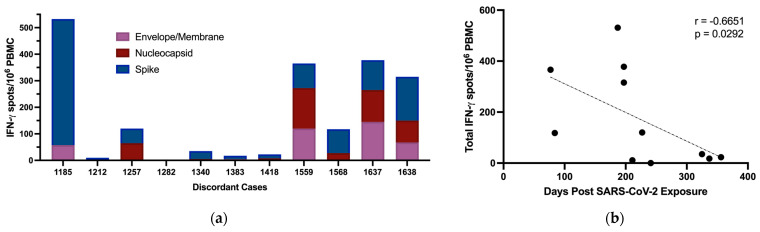
Production of IFN-γ by PBMC from discordant case subjects following stimulation with SARS-CoV-2 protein overlapping peptide pools. (**a**) IFN-γ producing T cells/10^6^ PBMC of discordant case subjects following 24 h stimulation with SARS-CoV-2 E/M, N, and S protein peptide pools. (**b**) Scatterplot with line of best fit showing the relationship between total IFN-γ producing T cells detected by ELISpot following stimulation and the number of days post exposure. Spearman correlations were computed to assess the significance of correlations, with the correlation coefficient (r) and *p* value shown within the graph plot. (**c**) Days post exposure and (**d**) age were compared between responders and non-responders on ELISpot. Red and blue shading of dots represents SARS-CoV-2 responders and non-responders respectively as defined by ELISPOT testing. The probability of a significant difference between groups was calculated using the Mann–Whitney test, with *p* values and ns (not significant) shown above lines spanning the groups compared.

**Figure 5 viruses-15-00996-f005:**
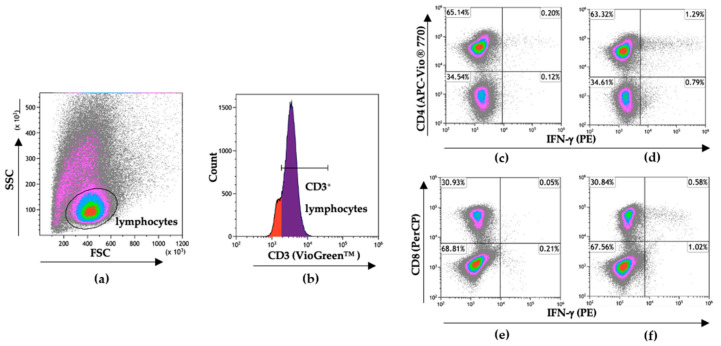
Flow cytometry gating strategy to discriminate SARS-CoV-2-specific CD4^+^ and CD8^+^ T cell responses. Following 7-day stimulation, cells were stained for extracellular CD3, CD4, and CD8 to discriminate T cell populations and for intracellular IFN-γ to identify SARS-CoV-2-specific T cells. (**a**) Gating on lymphocytes following 7-day in vitro stimulation. (**b**) Gating on CD3^+^ lymphocytes. (**c**) Non-restimulated CD4^+^IFN-γ^+^ cells and (**d**) CD4^+^ IFN-γ^+^ cells after 5 h stimulation with SARS-CoV-2 N protein peptide pool. (**e**) Non-restimulated CD8^+^IFN-γ^+^ cells and (**f**) CD8^+^IFN-γ^+^ cells after 5 h stimulation with SARS-CoV-2 S protein peptide pool. Data were analyzed and visualized using Kaluza Version 2.1 (Beckman Coulter). Total percentages of the (**g**) CD4^+^ and (**h**) CD8^+^ T cell populations responding to SARS-CoV-2 protein peptide pools with IFN-γ production after 7-day in vitro stimulation.

**Table 1 viruses-15-00996-t001:** Demographics of discordant cohort and details on exposure to SARS-CoV-2.

Subject ID	Sex	Age	Date of Contact’s Confirmatory Test	Contact’s Symptoms	SARS-CoV-2 Exposure Strain	Days from Exposure to Sample Collection
**1185**	F	68	25 March 2020	Severe	Wuhan-Hu-1	187
**1212**	M	63	30 March 2020	Moderate	Wuhan-Hu-1	211
**1257**	M	26	31 March 2020 to 4 April 2020 ^1^	Moderate (×1) Mild (×3) ^1^	Wuhan-Hu-1	227
**1282**	M	76	3 April 2020	Moderate	Wuhan-Hu-1	241
**1340**	F	56	15 March 2020	Severe	Wuhan-Hu-1	325
**1383**	F	54	31 March 2020	Moderate	Wuhan-Hu-1	337
**1418**	F	66	24 March 2020	Moderate	Wuhan-Hu-1	356
**1559**	F	57	10 February 2021 ^1^	Moderate (×2) ^1^	B.1.1.7	77
**1568**	M	49	11 February 2021	Moderate	B.1.1.7	84
**1637**	F	39	15 February 2021 ^2^	Mild ^2^	B.1.1.7	197
**1638**	M	68			B.1.1.7	197

^1^—Multiple family members tested positive over multiple days for SARS-CoV-2 infection with varying symptoms. ^2^—Exposed to the same family member.

## Data Availability

Data supporting the findings of this study and preserving anonymity of study participants are available from the corresponding author through electronic correspondence for legitimate scientific purposes.

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
