# Peer review of "Cellular Immune Responses to SARS-CoV-2 in Exposed Seronegative Individuals"

_viruses, 2023, doi:10.3390/v15040996_

Round 1
Reviewer 1 Report
The manuscript is on an interesting topic and overall well-written.
However, the very small sample size is a major concern which challenges the robustness of the presented data. Ideally, I would recommend to provide a formal sample size calculation based on one of the major/primary outcomes of the study, and accordingly, either confirm this size is appropriate, or if needed, increase the “n”. But, I understand it might not be possible due to the limit of access to new specimens which correspond to the initial design. In that case, the authors should clearly explain in the methods section (study population) how and why did they arrive to such a small sample size. Please clarify if there was an initial statistical sample size calculation, or, explain how many individuals were initially screened to get a maximum of n=11 for this study.
Another concern is the significant and long variation between the time of exposure and blood collection between 3 months to a year. This might drastically affect the observations and data interpretation, since it is well-documented that cellular / humoral immune responses to corona viruses decline overtime. This aspect should be considered in data interpretation and should be discussed in the discussion. Furthermore, please provide a new table which includes the r and p values of a spearman correlation test between all immunological valuables reported in the paper and the time of sampling to assess how such a delay between the exposure and sampling might affect the immune responses.
I also have concerns about the statistical analysis used in the manuscript and recommend to revise and confirm statistical analysis as follow:
· Line 217: please describe clearly which statistical tests were used in this manuscript. This information is missing in the methods.
· The statistical analysis should be revised and confirmed in all figures and the tests used in each figure, should be mentioned in their legends.
· Line 220, statistical analysis: please transform the table describing the asterisk(s) into a sentence whitin the text. There is no need for this table.
· Figure 1: it is unclear why the ANOVA test has been used to compare two variables. Furthermore, I think a Kruskal-Wallis test is more appropriate for such a small sample size, unless a distribution test confirms that ANOVA is more appropriate. In principal, the ANOVA test could be applied only in the figure 1d, but should be followed by a Mann Whitney test.
· The statistic asterisk(s) are missing in the figure 1d. please also indicate NS if there is no statistical differences.
· Figure 3: the Kruskal-Wallis test should be followed by a Wilcoxon signed-rank test to compare the matched (paired) variables.
· Figure 4: please clarify which correlation test was used for the figure 4d? as mentioned before, a spearman test should be provided for the correlation between all immunological outcomes, not only the IFNg production.
Other comments:
· Table 1 needs to be revised and includes the “unexposed individuals” which are used in the figures. The information about these unexposed individuals is missing.
· Please also confirm that during the period of the exposure and specimen collection, the individuals were not diagnosed by COVID-19 or other corona viruses infection or being vaccinated.
· Line 229: please describe what is the definition of “prolonged close contact” (How many days, etc)
· Please provide a flow cytometry gating strategy (including the control(s)) used to generate the data on IFN-g production.
· The discussion is missing the study limits section. Please provide an independent paragraph describing the study limits and challenges.
· There are also variations between the ages of the individuals (rang: 26-76 years). Can you please comment how the age might affect the results and if there is any association with the age.
Author Response
Thank you for your careful review and helpful comments on statistical tests used for analysis and other matters. We feel that addressing these comments has improved the clarity and overall strength of our manuscript. Changed and additional text is underlined.
The manuscript is on an interesting topic and overall well-written.
However, the very small sample size is a major concern which challenges the robustness of the presented data. Ideally, I would recommend to provide a formal sample size calculation based on one of the major/primary outcomes of the study, and accordingly, either confirm this size is appropriate, or if needed, increase the “n”. But, I understand it might not be possible due to the limit of access to new specimens which correspond to the initial design. In that case, the authors should clearly explain in the methods section (study population) how and why did they arrive to such a small sample size. Please clarify if there was an initial statistical sample size calculation, or, explain how many individuals were initially screened to get a maximum of n=11 for this study.
Unfortunately, it is not possible for us to increase the n based on our initial study design at this time because of the high level of vaccination and Omicron infection in the general population. We have added some information in the methods section on how we arrived at this n, including the total number of uninfected persons screened (Lines 78, 83-91).
Another concern is the significant and long variation between the time of exposure and blood collection between 3 months to a year. This might drastically affect the observations and data interpretation, since it is well-documented that cellular / humoral immune responses to corona viruses decline overtime. This aspect should be considered in data interpretation and should be discussed in the discussion. Furthermore, please provide a new table which includes the r and p values of a spearman correlation test between all immunological valuables reported in the paper and the time of sampling to assess how such a delay between the exposure and sampling might affect the immune responses.
We have added discussion on how decay in the cellular/humoral immune responses to coronaviruses over time could affect our data and interpretations (Lines 517-524). Spearman testing for correlation between time of sampling and immunological variables showed no significant correlation other than with SARS-CoV-2 peptide-induced IFNg spots/106 PBMC as shown in figure 4b. Therefore, we now explicitly state this in the results section (Lines 374-376), but did not add another table for these data.
I also have concerns about the statistical analysis used in the manuscript and recommend to revise and confirm statistical analysis as follow:
- Line 217: please describe clearly which statistical tests were used in this manuscript. This information is missing in the methods.
- The statistical analysis should be revised and confirmed in all figures and the tests used in each figure, should be mentioned in their legends.
Statistical tests used for analysis are now listed in the methods section (Lines 230-233) and specified in the appropriate figure captions.
- Line 220, statistical analysis: please transform the table describing the asterisk(s) into a sentence within the text. There is no need for this table.
This table has been deleted and p values indicated numerically on all graphs where appropriate.
- Figure 1: it is unclear why the ANOVA test has been used to compare two variables. Furthermore, I think a Kruskal-Wallis test is more appropriate for such a small sample size, unless a distribution test confirms that ANOVA is more appropriate. In principal, the ANOVA test could be applied only in the figure 1d, but should be followed by a Mann Whitney test.
Non-parametric Mann Whitney testing for differences has now been carried out as suggested and noted in the figure caption.
- The statistic asterisk(s) are missing in the figure 1d. please also indicate NS if there is no statistical differences.
- Figure 3: the Kruskal-Wallis test should be followed by a Wilcoxon signed-rank test to compare the matched (paired) variables.
The Wilcoxon signed-rank test was used to compare the matched (paired) variables as suggested with p values shown above the groups compared as noted in the figure caption.
- Figure 4: please clarify which correlation test was used for the figure 4d? as mentioned before, a spearman test should be provided for the correlation between all immunological outcomes, not only the IFNg production.
We have clarified in the figure caption (now figure 4b) that a spearman test was done to calculate correlation between IFNg production and time of sample collection. Spearman testing was done for correlation between time of sample collection and all other immunological variables with no significant correlation found, as mentioned in the results section (Lines 374-376).
Other comments:
- Table 1 needs to be revised and includes the “unexposed individuals” which are used in the figures. The information about these unexposed individuals is missing.
Matching for age and sex was done as described (Lines 319-320), but the pre-pandemic plasma samples are archived from previous studies and, thus, were anonymized for their use as pre-pandemic controls in this study.
- Please also confirm that during the period of the exposure and specimen collection, the individuals were not diagnosed by COVID-19 or other corona viruses infection or being vaccinated.
This point is confirmed in the methods section (Lines 247-249).
- Line 229: please describe what is the definition of “prolonged close contact” (How many days, etc)
More information on what is defined as prolonged close contact is provided in the methods section (Lines 85-88 ).
- Please provide a flow cytometry gating strategy (including the control(s)) used to generate the data on IFN-g production.
The gating strategy for measuring IFNg production is now shown in figure 5a-f.
The discussion is missing the study limits section. Please provide an independent paragraph describing the study limits and challenges.
A separate paragraph has been added to the discussion addressing study limits and challenges (Lines 577-587).
- There are also variations between the ages of the individuals (rang: 26-76 years). Can you please comment how the age might affect the results and if there is any association with age.
We added brief mention to how age could potentially affect the results (Lines 376-379) and tested for association between age and detection of cellular immune response against SARS-CoV-2 (Figure 4d).
Reviewer 2 Report
In this article the authors wanted to characterise immunity against SARS-CoV-2 in eleven individuals who were negative by nucleic acid testing during prolonged close contact and who had no serological diagnosis of infection. The authors state that this situation could have arisen due to a number of issues including natural immunity; cross-reactive immunity from previous coronavirus exposure; abortive infection due to de novo immune responses or a variety of other factors.
The authors point out that exposed uninfected individuals were seronegative against SARS-CoV-2 spike and also selectively reactive against OC43 nucleocapsid protein, suggesting common beta-coronavirus exposure induced antibody cross-reactive against SARS-CoV-2 nucleocapsid protein. There was no evidence of protection from any form of circulating angiotensin-converting enzyme or IFN-alpha. Six individuals had T-cell responses against SARS-CoV-2, four involving CD4+ and CD8+ T-cells. The authors found no evidence of protection from SARS-CoV-2 through innate immunity or immunity induced by common beta-coronaviruses.
Some of the individuals in the study had cellular immune responses against SARS-CoV-2
associated with their time since exposure and so the authors suggest that rapid cellular responses could contain SARS-CoV-2 infection below thresholds required for a humoral response.
The authors highlight that potentially over half of the Canadian population has been infected with SARS-CoV-2 and that research carried out over the last three years has helped to inform public health agencies on best practice to assist in addressing issues related to the pandemic. SARS-CoV-2 is no different from other viruses in that exposure to it can sometimes occur without any overt clinical signs of infection and also without any seroconversion and where some cases display detectable cellular immunity against SARS-CoV-2. The authors go on to suggest that other published data support the idea that in some rare cases, viral infections can be cut short ahead of seroconversion by pre-existing or rapidly developing cellular immunity.
The authors note that fairly recent published data demonstrates that infection with a common coronavirus before any infection with SARS-CoV-2 can lessen COVID-19 disease severity. Based on a number of publications, it is very unclear however whether cross-reactive antibodies or other forms of immunity induced by infection with common coronaviruses provide protection against SARS-CoV-2 infection or against severe COVID-19.
In the present study, the authors assessed the immune responses against SARS-CoV-2 of individuals who were in prolonged, close contact to an active case of COVID-19 but appeared to remain in an uninfected state. The study individuals showed no evidence of viral replication by reverse transcriptase polymerase chain reaction testing. They had no symptoms of clinical disease and remained seronegative against the immunodominant SARS-CoV-2 spike protein.
The authors conclude that there is probably major individual variability in the nature of exposure and so many cases will be affected differently and therefore have a different outcome and different presentation.
Main points and comments:
- The manuscript has been well written, clearly and concisely presented and the authors have attempted to explain the data they have generated. It is an interesting read and the manuscript delivers exactly what it says it will in the title.
- My main concern is that the detection of T cell responses against SARS-CoV-2 varies very widely in the seronegative population depending on the method used and the potential number of T cell epitopes needs to be expanded in order to give more robust and accurate data analysis when looking at incidence of induction of SARS-CoV-2 cellular immunity from exposure but without any form of seroconversion. I agree with the authors that T cell responses from exposure without seroconversion will be less robust than those caused by a severe infection.
- One other issue with this manuscript is the extremely low number of participants in this particular study. I realise that this is the nature of the study, but the ‘n’ value is small. I know there will be other similar cases elsewhere and I do know it is difficult to carry out this type of work, but can the authors please comment on this, and can they say how confident they are with the results they have? As the authors state themselves – there have been over 662 million known cases of COVID-19 as of January 2023 and they are currently assessing 11 individuals.
- Can the authors obtain access to any other relevant samples from other studies that may contain individuals of a similar status as those tested here? Again, I know this is not easy.
- The objective set out by the authors has certainly been fulfilled in this manuscript and it does what the title says, but I am not convinced the data will give particularly robust results as so much of this is “unclear” or “unknown” due to the nature of the exercise. No real conclusions can be drawn from the data presented here.
- The authors have presented ethically sound research.
- The authors have assessed a variety of assays and the results are clear-cut in the sense that they are “clean” although not always as expected.
- All of the assays used are relevant and suitable for this type of work.
- There are so many things that are outside of the control of the authors such as the days post exposure in Figure 4b; very little correlation between days post SARS-CoV-2 exposure and total IFN-gamma spots/10*6 PBMC in Figure 4c.
- Personally, I would find it helpful to have the statistical analyses added to the end of each of the relevant legends rather than just in the Materials and Methods as they are currently. Please can the relevant stats be added to Figure 1, Figure 3 and Figure 4 in the form of a p value? Figure 2 mentions statistical analyses but no asterisks are shown. Surely Figure 2a should be highly significant? Can this be clarified please with either the stats shown on the figure itself and in the legend or a comment added to the end of the legend stating there were no statistically significant differences detected in the analyses? If stats were not carried out on Figure 2, please adjust accordingly.
- Figure 3 shows OD450nm values of less than ‘1’ so even though the axes are quite large, the values shown are relatively low. Is this the expected dynamic range of this assay? Are there any positive controls for the assay itself to give an indication of the dynamic range expected?
- Figure 3 is not clear with regards to which groups are statistically significant. ie are the lines above the plots referring to all 3 groups in each set being compared with each other? Are there any analyses between the discordant results and the pre-pandemic controls as opposed to the stats being within each group?
- Figure 4a does appear to be random distribution of data. Can the authors please comment?
- I appreciate it is difficult to obtain large enough volumes of sample to fully investigate all avenues, but this manuscript does give some insight into some very unpredictable data sets.
- Can the authors suggest what percentage of the population they think fit into the category they are testing in this manuscript? What is the significance of these individuals? Can they extrapolate from the 11 individuals to the population as a whole?
- Can the authors comment on whether most of the population could, or should, have been exposed to at least one common beta-coronavirus in their lifetime (regardless of age, sex or location)?
- Can the authors comment about the suspected number of asymptomatic COVID-19 cases?
- Do the authors think the results they have generated are really looking at cross-reactivity?
- Minor point: Line 517 has 2 extra words that require deleting (“…..showed that showed that…..”).
- The references that have been cited are relevant and suitable.
Author Response
We thank you for your comprehensive review and helpful comments. Addressing them has improved the quality of our manuscript. Changed and additional is underlined.
In this article the authors wanted to characterise immunity against SARS-CoV-2 in eleven individuals who were negative by nucleic acid testing during prolonged close contact and who had no serological diagnosis of infection. The authors state that this situation could have arisen due to a number of issues including natural immunity; cross-reactive immunity from previous coronavirus exposure; abortive infection due to de novo immune responses or a variety of other factors.
The authors point out that exposed uninfected individuals were seronegative against SARS-CoV-2 spike and also selectively reactive against OC43 nucleocapsid protein, suggesting common beta-coronavirus exposure induced antibody cross-reactive against SARS-CoV-2 nucleocapsid protein. There was no evidence of protection from any form of circulating angiotensin-converting enzyme or IFN-alpha. Six individuals had T-cell responses against SARS-CoV-2, four involving CD4+ and CD8+ T-cells. The authors found no evidence of protection from SARS-CoV-2 through innate immunity or immunity induced by common beta-coronaviruses.
Some of the individuals in the study had cellular immune responses against SARS-CoV-2
associated with their time since exposure and so the authors suggest that rapid cellular responses could contain SARS-CoV-2 infection below thresholds required for a humoral response.
The authors highlight that potentially over half of the Canadian population has been infected with SARS-CoV-2 and that research carried out over the last three years has helped to inform public health agencies on best practice to assist in addressing issues related to the pandemic. SARS-CoV-2 is no different from other viruses in that exposure to it can sometimes occur without any overt clinical signs of infection and also without any seroconversion and where some cases display detectable cellular immunity against SARS-CoV-2. The authors go on to suggest that other published data support the idea that in some rare cases, viral infections can be cut short ahead of seroconversion by pre-existing or rapidly developing cellular immunity.
The authors note that fairly recent published data demonstrates that infection with a common coronavirus before any infection with SARS-CoV-2 can lessen COVID-19 disease severity. Based on a number of publications, it is very unclear however whether cross-reactive antibodies or other forms of immunity induced by infection with common coronaviruses provide protection against SARS-CoV-2 infection or against severe COVID-19.
In the present study, the authors assessed the immune responses against SARS-CoV-2 of individuals who were in prolonged, close contact to an active case of COVID-19 but appeared to remain in an uninfected state. The study individuals showed no evidence of viral replication by reverse transcriptase polymerase chain reaction testing. They had no symptoms of clinical disease and remained seronegative against the immunodominant SARS-CoV-2 spike protein.
The authors conclude that there is probably major individual variability in the nature of exposure and so many cases will be affected differently and therefore have a different outcome and different presentation.
Main points and comments:
- The manuscript has been well written, clearly and concisely presented and the authors have attempted to explain the data they have generated. It is an interesting read and the manuscript delivers exactly what it says it will in the title.
Thank you for your generous comments.
- My main concern is that the detection of T cell responses against SARS-CoV-2 varies very widely in the seronegative population depending on the method used and the potential number of T cell epitopes needs to be expanded in order to give more robust and accurate data analysis when looking at incidence of induction of SARS-CoV-2 cellular immunity from exposure but without any form of seroconversion. I agree with the authors that T cell responses from exposure without seroconversion will be less robust than those caused by a severe infection.
We agree with the reviewer that a more complete representation of the SARS-CoV-2 peptidome is needed for truly definitive studies of the incidence of induction of SARS-CoV-2 cellular immunity in the absence of seroconversion. We were limited by both the availability of appropriate samples, PBMC numbers and cost/availability of SARS-CoV-2 peptides. These factors are now mentioned in the paragraph added on study challenges and limitations (Lines 577-587).
- One other issue with this manuscript is the extremely low number of participants in this particular study. I realise that this is the nature of the study, but the ‘n’ value is small. I know there will be other similar cases elsewhere and I do know it is difficult to carry out this type of work, but can the authors please comment on this, and can they say how confident they are with the results they have? As the authors state themselves – there have been over 662 million known cases of COVID-19 as of January 2023 and they are currently assessing 11 individuals.
We are confident in the material results we obtained studying these 11 individuals, however, our interpretation is speculative and the small n is discussed (Lines 552-553, 577-587) as a limitation to any potential extrapolation of results to the general population.
- Can the authors obtain access to any other relevant samples from other studies that may contain individuals of a similar status as those tested here? Again, I know this is not easy.
We don’t know of any existing samples collected prior to the introduction of vaccines and widespread Omicron infections that we can access. Ethical implications of obtaining samples from another site would prevent generation of any related data in a timely fashion.
- The objective set out by the authors has certainly been fulfilled in this manuscript and it does what the title says, but I am not convinced the data will give particularly robust results as so much of this is “unclear” or “unknown” due to the nature of the exercise. No real conclusions can be drawn from the data presented here.
We agree with the reviewer that no real conclusions can be drawn beyond the individual findings themselves. Corroboration of the phenomenon of T cell responses against SARS-CoV-2 developing in the absence of overt infection together with reference to similar findings with other viruses is of value to stimulate further research in an area relevant to the pathogenesis of viral infection and vaccine design.
- The authors have presented ethically sound research.
- The authors have assessed a variety of assays and the results are clear-cut in the sense that they are “clean” although not always as expected.
- All of the assays used are relevant and suitable for this type of work.
Thank you for these positive evaluations.
- There are so many things that are outside of the control of the authors such as the days post exposure in Figure 4b; very little correlation between days post SARS-CoV-2 exposure and total IFN-gamma spots/10*6 PBMC in Figure 4c.
We have redone the correlation analysis using the non-parametric spearman test appropriate for our data and now report an significant correlation between days post SARS-CoV-2 exposure and total IFN-gamma spots/10*6 PBMC (Figure 4b). This is the only immunological variable that correlates with days post SARS-CoV-2 exposure as now stated in the results section (Lines 374-376).
- Personally, I would find it helpful to have the statistical analyses added to the end of each of the relevant legends rather than just in the Materials and Methods as they are currently. Please can the relevant stats be added to Figure 1, Figure 3 and Figure 4 in the form of ap value? Figure 2 mentions statistical analyses but no asterisks are shown. Surely Figure 2a should be highly significant? Can this be clarified please with either the stats shown on the figure itself and in the legend or a comment added to the end of the legend stating there were no statistically significant differences detected in the analyses? If stats were not carried out on Figure 2, please adjust accordingly.
We have added description of the statistical tests used for analysis/comparison in the relevant figure legends as suggested. Figure 2a was not meant to compare the group of discordant subjects with infected samples. We included a positive control to demonstrate that the assay does successfully measure interference between SARS-CoV-2 spike and ACE2 binding when such interference is present. Data points from several additional samples with inhibitory activity run under the same conditions has been added, but a statistical comparison would be meaningful.
- Figure 3 shows OD450nm values of less than ‘1’ so even though the axes are quite large, the values shown are relatively low. Is this the expected dynamic range of this assay? Are there any positive controls for the assay itself to give an indication of the dynamic range expected?
We developed a standardized assay to measure IgG antibodies against the different coronavirus N proteins to detect infections occurring post-vaccination and to distinguish infection with SARS-CoV-2 from infection with common coronaviruses. Some recently infected individuals have OD450 values between 2 and 3 using this assay as described in a previous publication (reference 1).
- Figure 3 is not clear with regards to which groups are statistically significant. ie are the lines above the plots referring to all 3 groups in each set being compared with each other? Are there any analyses between the discordant results and the pre-pandemic controls as opposed to the stats being within each group?
We have clarified which data sets have been compared with individual bars spanning the groups and p values above. We did compare results between the discordant subjects and pre-pandemic controls and now show with lines above the groups compared that there was no significant difference in ODs against SARS-CoV-2 or either of the common b-coronaviruses.
- Figure 4a does appear to be random distribution of data. Can the authors please comment?
In figure 4a, the coloured segments of different bars represent the number of PBMC/106 for each of the numbered subjects producing IFNg in response to each of 3 different pools of SARS-CoV-2 peptides.
- I appreciate it is difficult to obtain large enough volumes of sample to fully investigate all avenues, but this manuscript does give some insight into some very unpredictable data sets.
There is a lot of individual variability in these types of studies.
- Can the authors suggest what percentage of the population they think fit into the category they are testing in this manuscript? What is the significance of these individuals? Can they extrapolate from the 11 individuals to the population as a whole?
We can speculate as to what percentage of the population fit into this category at the time the samples were collected, which is probably much different now, but worth discussing (Lines 567-570).
- Can the authors comment on whether most of the population could, or should, have been exposed to at least one common beta-coronavirus in their lifetime (regardless of age, sex or location)?
My understanding is that (virtually) everyone is exposed to a common b-coronavirus in their lifetime, with more frequent exposure as children (comment added to discussion (Lines 495-498).
- Can the authors comment about the suspected number of asymptomatic COVID-19 cases?
We can speculate on the suspected number of asymptomatic COVID-19 cases being around 30%, but those cases would be defined by a positive PCR test and therefore, differ from the asymptomatic cases we identified as our discordant cohort.
- Do the authors think the results they have generated are really looking at cross-reactivity?
We are confident that the low level of antibodies against SARS-CoV-2 N protein do represent cross-reactivity and speculate with reason that the T cell responses we observe against SARS-CoV-2 arose from exposure to SARS-CoV-2 (Lines 548-552).
- Minor point: Line 517 has 2 extra words that require deleting (“…..showed that showed that…..”).
Thank you for pointing this out. We have corrected the duplication.
- The references that have been cited are relevant and suitable.
Reviewer 3 Report
The work by Norton et al. describes the immune landscape of a cohort of 11 individuals that were asymptomatic after prolonged continuous exposure to SARS-CoV-2 and did not test positive by standard quantitative PCR assay. The contact’s had ranging severity of infection after confirmatory positive testing. The “discordant” cohort were assessed for serology using ELISA, innate immunity by INFa, cross reactivity to other pan-coronaviruses, and production of IFNy by ELISPOT and flow cytometry. The work is nicely detailed and very interesting, however, could be further enhanced by addressing the comments below.
Comments:
Line 233: Please detail how the “discordant” cohort was determined to be asymptomatic, were these individuals monitored over the period of exposure and prior to sample collection?
Some figures in the manuscript have different quality compared to others i.e. figure 1, panel b, c, and d are low quality compared to panel a.
Figure 1:
Did the authors consider ELISA end point titers or alternative quantitative methods for titers compared to raw OD450 for reporting? Otherwise, the comparisons of post infection and discordant samples cannot be said to be seronegative, certainly there is a significant difference between the samples in terms of “high” and “low” antibody titers, however as seen in panel a there may be some “low seropositive” individuals in the discordant group for FLS compared to RBD. Alternatively known seronegative control individuals can be included in the figure to strengthen the author’s statement of seronegativity.
Panel b and c seem to only include one positive control, for comparison more positive controls should be included particularly for statistical significance determination.
Panel d has no asterisks above figures for statistical determination. Please include details on which discordant samples were run since the entire cohort was not included.
Figure 2:
Panel a, please include rationale for only including 1 positive control. If possible, would be better to include more.
Figure 3:
This panel re-affirms my first comment on figure 1 since it looks like there are low levels of anti-SARS-CoV-2 N antibodies, therefore the discordant individuals may not be “true seronegative”.
Figure 5:
Can the authors describe a reasoning behind not detecting IFNy positive T cells from 1257 and 1559 even though there was a production in IFNy by ELISPOT from figure 4?
Please describe reasoning for not including the remaining 5 individuals from the discordant group? How did their T cell responses compare?
Author Response
We thank you for your careful review and helpful comments. They have helped to improve the quality of our manuscript. Changed and additional text is underlined.
The work by Norton et al. describes the immune landscape of a cohort of 11 individuals that were asymptomatic after prolonged continuous exposure to SARS-CoV-2 and did not test positive by standard quantitative PCR assay. The contact’s had ranging severity of infection after confirmatory positive testing. The “discordant” cohort were assessed for serology using ELISA, innate immunity by INFa, cross reactivity to other pan-coronaviruses, and production of IFNy by ELISPOT and flow cytometry. The work is nicely detailed and very interesting, however, could be further enhanced by addressing the comments below.
Comments:
Line 233: Please detail how the “discordant” cohort was determined to be asymptomatic, were these individuals monitored over the period of exposure and prior to sample collection?
Subjects completed a questionnaire upon study entry wherein they self-reported an absence of symptoms coincident with the time period immediately preceding and following a documented negative RT-PCR indicated by their close contact to a person or persons with confirmed COVID-19 (Lines 81-82, 243-246).
Some figures in the manuscript have different quality compared to others i.e. figure 1, panel b, c, and d are low quality compared to panel a.
Image quality for all figures has been increased to a minimum 600 dpi.
Figure 1:
Did the authors consider ELISA end point titers or alternative quantitative methods for titers compared to raw OD450 for reporting? Otherwise, the comparisons of post infection and discordant samples cannot be said to be seronegative, certainly there is a significant difference between the samples in terms of “high” and “low” antibody titers, however as seen in panel a there may be some “low seropositive” individuals in the discordant group for FLS compared to RBD. Alternatively known seronegative control individuals can be included in the figure to strengthen the author’s statement of seronegativity.
We appreciate the reviewer’s point that the definition of seronegativity can be affected by the sample dilution chosen for ELISA measurements, but didn’t consider measuring end point titers to complement raw OD450s. We previously established an objective cut-off for ELISA OD for anti-SARS-CoV-2 spike reactivity at this plasma dilution with a large set of pre-pandemic plasma samples as negative controls (reference 1).
Panel b and c seem to only include one positive control, for comparison more positive controls should be included particularly for statistical significance determination.
In these cases, we were not trying to compare data from the discordant subjects with another group, but included a positive control to show that the assay does work. Additional positive controls have been added to the figures, but no statistical comparison was done.
Panel d has no asterisks above figures for statistical determination. Please include details on which discordant samples were run since the entire cohort was not included.
We have clarified the p values calculated by placing them above bars spanning the groups compared and described the method of statistical analysis in the figure caption. The discordant subjects we had post vaccine 1 samples from are now listed in the figure caption by the same IDs as in Table 1.
Figure 2:
Panel a, please include rationale for only including 1 positive control. If possible, would be better to include more.
We were not trying to compare data from the discordant subjects with another group, but included a positive control to show that the assay does work. Additional positive controls have been added to the figures.
Figure 3:
This panel re-affirms my first comment on figure 1 since it looks like there are low levels of anti-SARS-CoV-2 N antibodies, therefore the discordant individuals may not be “true seronegative”.
As pointed out by the reviewer, there are low levels of anti-SARS-CoV-2 N antibodies detected. We surmise that this is due to cross-reactivity with the common b-coronaviruses the discordant subjects were previously exposed to as the reactivity in every case is selective for the N protein of OC43. This was addressed in a previous publication (reference 1) where the hierarchy of reactivity against different b-coronavirus N proteins was used to distinguish cross-reactivity from exposure to common b-coronaviruses from SARS-CoV-2 infection.
Figure 5:
Can the authors describe a reasoning behind not detecting IFNy positive T cells from 1257 and 1559 even though there was a production in IFNy by ELISPOT from figure 4?
We speculate that the cells in PBMC producing IFNg by ELISPOT for subjects 1257 and 1559 were not stable SARS-CoV-2-specific memory cells that could be expanded in vitro or were producing IFNg through collateral activation by T cells responding against non-SARS-CoV-2 antigens. It was previously reported that collateral activation renders T cells susceptible to apoptosis with longer term in vitro stimulation (doi.org/10.4049/jimmunol.0802596). These possibilities are presented in the discussion (Lines 532-536).
Please describe reasoning for not including the remaining 5 individuals from the discordant group? How did their T cell responses compare?
We felt that with no SARS-CoV-2 specific T cells being detected in PBMC, the likelihood of expanding SARS-CoV-2 specific T cells with in vitro stimulation was negligible.
Reviewer 4 Report
In this study, Norton et al. evaluate cellular immune responses to SARS-CoV-2 in exposed seronegative individuals. For that, they use eleven individuals who after a prolonged close contact to an active case of COVID-19 remained apparently uninfected with a negative nucleic acid test and no serological diagnosis of infection.
The manuscript is well-written and well-structured but the sample size used for this analysis is small (n=11). Considering that half of the individuals are responders (n=6) the conclusions suggested by the authors are too speculative.
Some comments and suggestions could be used by the authors to improve their manuscript:
-Introduction
Lines 66-70: Please check the font size of the last paragraph, it is different from the rest. I'm not sure if it occurred during the conversion to pdf or it was like that in the original manuscript
-Results
SARS-CoV-2 serology
The first sentence needs to be revised; some infected individuals are SARS-CoV-2 antibody negative (also named non-seroconverters) nonetheless have detectable SARS-CoV-2 specific T cell responses. This phenomenon is widely described in literature, especially in Long Covid patients.
Figure 1: Why do the authors use post infection samples versus discordant samples for Ig G antibody responses (a and d) but a positive control versus discordant samples for IgM and IgA (b and c)? The reason should be mention, otherwise the data comparing post infection samples vs discordant samples must be included.
Cross-reactive immunity with common b-coronaviruses
Figure 3. This graph lacks post-infection samples that would be very useful to confirm the cross-reactivity of SARS-CoV-2 with HKU1 and OC43.
T-cell responses to SARS-CoV-2
The authors have to define ELISpot responders and non-responders. For example, in line 366: …to at least one of the peptide pools (responders). Or something similar but it should be mentioned.
Author Response
We thank you for your careful review and helpful comments. They have helped to improve the quality of our manuscript. Changed and additional text is underlined.
In this study, Norton et al. evaluate cellular immune responses to SARS-CoV-2 in exposed seronegative individuals. For that, they use eleven individuals who after a prolonged close contact to an active case of COVID-19 remained apparently uninfected with a negative nucleic acid test and no serological diagnosis of infection.
The manuscript is well-written and well-structured but the sample size used for this analysis is small (n=11). Considering that half of the individuals are responders (n=6) the conclusions suggested by the authors are too speculative.
The conclusions we arrive at based on results for 11 individuals are speculative and can only be applied in these particular instances, not extrapolated to a larger population. We have commented in the paragraph on study limitations (Line 577-587).
Some comments and suggestions could be used by the authors to improve their manuscript:
-Introduction
Lines 66-70: Please check the font size of the last paragraph, it is different from the rest. I'm not sure if it occurred during the conversion to pdf or it was like that in the original manuscript
Thank you for noticing this issue. It has been corrected.
-Results
SARS-CoV-2 serology
The first sentence needs to be revised; some infected individuals are SARS-CoV-2 antibody negative (also named non-seroconverters) nonetheless have detectable SARS-CoV-2 specific T cell responses. This phenomenon is widely described in literature, especially in Long Covid patients.
We found previously only one case out of 44 individuals with a PCR confirmed COVID-19 test who remained seronegative, but are aware of literature suggesting that mild or asymptomatic infection can occur with out seroconversion in a substantial fraction of cases. The individuals we selected for study did not have a positive PCR test for SARS-CoV-2 infection and we looked at serology for confirmation here were no overt signs of infection. We have changed the first sentence of this section to include the idea that SARS-CoV-2 infection can occur in the absence of seroconversion:
Figure 1: Why do the authors use post infection samples versus discordant samples for IgG antibody responses (a and d) but a positive control versus discordant samples for IgM and IgA (b and c)? The reason should be mention, otherwise the data comparing post infection samples vs discordant samples must be included.
In figure 1, we labelled the positive controls for IgG as post-infection samples because the samples were from persons with known infection. For IgA, IgM and pseudoneutralization, we chose several samples we previously knew to be positive to demonstrate that the assay worked to measure these activities and that the discordant subjects were negative when tested at the same dilution as the positive controls. There was no between group comparison intended. We have changed the labelling in every case to positive controls for consistency.
Cross-reactive immunity with common b-coronaviruses
Figure 3. This graph lacks post-infection samples that would be very useful to confirm the cross-reactivity of SARS-CoV-2 with HKU1 and OC43.
Cross-reactivity between SARS-CoV-2, HKU1 and OC43 N proteins has previously been demonstrated and illustrated with post SARS-CoV-2 infection samples (Reference 1).
T-cell responses to SARS-CoV-2
The authors have to define ELISpot responders and non-responders. For example, in line 366: …to at least one of the peptide pools (responders). Or something similar but it should be mentioned.
We have revised a statement in the methods section to clarify that subjects were considered ELISpot responders if they produced > 50 IFNg spots/106 PBMC above background in response to any of the SARS-CoV-2 peptide pools tested (Lines185-188).
Round 2
Reviewer 1 Report
the authors addressed my comments adequately
Author Response
Thank you for your comprehensive review.
Reviewer 3 Report
The authors have addressed my comments appropriately.
I would recommend including a description of the cut off for the pre pandemic controls per my first comment. The highlight of this manuscript is that these individuals are seronegative as shown in tested pre-pandemic controls.
Author Response
Thank you for the suggestion. We have added a description of how we established a cutoff for IgG seropositivity against SARS-CoV-2 S and RBD with pre-pandemic control samples in the methods section (lines 123-126) and put cut-off lines at the appropriate ODs on figure 1a.
Reviewer 4 Report
The authors have introduce the required modifications.
Author Response

(The authors gave the same response as above.)
